# Repair and Mechanism of Oligopeptide SEP-3 on Oxidative Stress Liver Injury Induced by Sleep Deprivation in Mice

**DOI:** 10.3390/md21030139

**Published:** 2023-02-22

**Authors:** Xin Hou, Chong Yi, Zekun Zhang, Hui Wen, Yufeng Sun, Jiaxin Xu, Hongyu Luo, Tao Yang

**Affiliations:** 1Key Laboratory of Health Risk Factors for Seafood of Zhejiang Province, Zhejiang Ocean University, Zhoushan 316022, China; 2Yantai Marine Economic Research Institute, Yantai 264003, China

**Keywords:** cyprinus oligopeptide, sleep deprivation, oxidative stress, biological rhythm

## Abstract

To investigate the effects of bonito oligopeptide SEP-3 on the repair of liver damage and regulation of liver biorhythm in sleep-deprived mice (SDM), C57BL/6 male mice were subjected to sleep deprivation by modified multi-platform water environment method, and were given different doses of bonito oligopeptide SEP-3 in groups. To determine the liver organ index, liver tissue-related apoptotic protein levels, Wnt/β-Catenin pathway-related protein expression levels, serum alanine transaminase (ALT), glutamicum transaminase (AST), glucocorticoid (GC), and adrenocorticotropin (ACTH) content in each group of mice, four time points were selected to examine the mRNA expression levels of circadian clock-related genes in mouse liver tissue. The results showed that low, medium, and high doses of SEP-3 significantly increased SDM, ALT, and AST (*p* < 0.05), and medium and high doses of SEP-3 significantly reduced SDM liver index and GC and ACTH. As SEP-3 increased the apoptotic protein and Wnt/β-Catenin pathway, mRNA expression gradually tended to normal (*p* < 0.05). This suggests that sleep deprivation can cause excessive oxidative stress in mice, which can lead to liver damage. Additionally, oligopeptide SEP-3 achieves the repair of liver damage by inhibiting SDM hepatocyte apoptosis, activating liver Wnt/β-Catenin pathway, and promoting hepatocyte proliferation and migration, and suggests that oligopeptide SEP-3 is closely related to repair of liver damage by regulating the biological rhythm of SDM disorder.

## 1. Introduction

The high pressure of modern society, rich nightlife, and Internet addiction had led to an increasing incidence of sleep disorders. It is reported that about 38.5% of Chinese people suffer from sleep disorders, and the number of women with sleep disorders is 1.5~2.0 times that of men [1]. Studies have shown that sleep deprivation can significantly increase the expression of 8-OHDG in the liver of the body, cause DNA disruption in hepatocytes, and then lead to apoptosis of hepatocytes, abnormal biochemical indicators such as alanine aminotransferase (ALT) and total bilirubin (TBIL), and impair cognitive and physical functions [2,3,4]. Li et al. [5] found that sleep deprivation not only induced oxidative stress response in rats, produced a large number of reactive oxygen species (ROS) to attack hepatocytes through mitochondrial consumption of oxygen, resulting in damage to hepatocytes, but also enhanced autophagy of hepatocytes by inhibiting the AKT/mTOR signaling pathway, thus leading to apoptosis of hepatocytes. Xing et al. [6] found that 72 h sleep deprivation can induce abnormal expression of various core clock genes in rat liver at both transcription and translation levels, and then lead to disorder of liver biological rhythm.

Some studies have shown that His-Gly-Pro-Hyp-Gly-Glu (TGP5), Asp-Gly-Pro- Lys-Gly-His (TGP7), and Met-Leu-Gly-Pro-Phe were obtained by enzymatic hydrolysis of bonito scale gelatin Gly-Pro-Ser (TGP9), and other oligopeptides showed a high scavenging ability of the DPPH radical, hydroxyl radical, and superoxide anion radical, indicating that the skipjack enzymatic peptide had strong antioxidant capacity in vitro [7]. Previous studies in our group showed that many peptides extracted from aquatic by-products have good antioxidant activity, such as GEYGFE, PSVSLT and IELFPGLP extracted from Siberian sturgeon, which reduce the content of reactive oxygen species (ROS) and malondialdehyde (MDA) by regulating the endogenous antioxidant enzymes of superoxide dismutase (SOD) and glutathione peroxidase (GSH-Px), showing significant cytoprotective effects on HUVECs [8]; YEGDP and WF extracted from blue mussels can protect HUVECs from H_2_O_2_ damage by increasing SOD, GSH-Px and NO levels, reducing ROS and malondialdehyde content [9]; Wang et al. [10] hydrolyzed oligopeptides such as GHHAAA, PHPR, SVTEV, VRDQY, and SMDV obtained from bonito spleen by enzymatic hydrolysis technology and acted on H_2_O_2_-induced HUVECs, and found that they could improve the activities of antioxidant enzymes (such as SOD) and GSH-Px in mice. The content of ROS and MDA was decreased.

The oligopeptide SEP-3 prepared by our research group using bonito waste showed that it could significantly improve the oxidative stress damage caused by sleep deprivation [8]. The repair level of liver injury is related to various physiological and biochemical indexes of liver. Therefore, in this paper, mice were deprived of sleep for 72 h by the modified multi-platform water environment method, and then SEP-3 was used for intervention. The viscera index, the contents of ALTAST, GC, and CTH in serum, the expression levels of apoptosis-related proteins and Wnt/β-Catenin pathway-related proteins in liver tissue, and the mRNA expression levels of clock-related genes in liver tissue were determined. The aim was to explore the repair effect and mechanism of skipjack oligopeptide SEP-3 on liver injury in sleep-deprived C57BL male mice.

## 2. Results

### 2.1. Effect of SEP-3 on Liver Injury Repair in Sleep-Deprived Mice

#### 2.1.1. Effects of SEP-3 on Liver Index in Sleep-Deprived Mice

To explore the effect of SEP-3 on liver damage caused by sleep deprivation, we measured the level of liver index after 72 h of sleep deprivation, as shown in Figure 1. Compared with the normal group, the liver index of SD group was significantly decreased (*p* < 0.05). The liver index of the three groups treated with SEP-3 was higher than that of the SD group, and increased in a dose-dependent manner, but there was no significant difference between the LSEP-3 (Low SEP-3, 20 mg/kg) group and the SD group (*p* > 0.05), and the LSEP-3 group was significantly different from the MSEP-3 (Middle SEP-3, 40 mg/kg) and HSEP-3 (High SEP-3, 80 mg/kg) groups (*p* < 0.05). However, there was no significant difference between the MSEP-3 group and HSEP-3 group (*p* > 0.05). Although the liver index of mice in each administration group was lower than that in normal group, there was no significant difference between the normal group and HSEP-3 group (*p* > 0.05). The above results showed that sleep deprivation for 72 h induced liver atrophy in mice, impaired liver function, and led to a significant reduction in liver index. Guo Xiaolei et al. [11] found that liver index was significantly reduced and liver atrophy occurred in rats with sleep deprivation, which was similar to the results of this study. However, high-dose SEP-3 administration can significantly improve the liver index of SDM and has a strong effect on liver tissue repair.

#### 2.1.2. Effect of SEP-3 on the Expression Levels of Serum ALT and AST in Sleep-Deprived Mice

The effects of SEP-3 on SDM liver function indexes were investigated by detecting the activities of ALT and AST in mouse serum, and the results were shown in Figure 2. Compared with the normal group, the activities of ALT and ASL in serum of the SD group were significantly increased (*p* < 0.05). The activities of serum ALT and AST in the LSEP-3, MSEP-3, and HSEP-3 groups were 19.19, 18.87, 18.06 U/L and 59.21, 47.43, 36.87 U/L, respectively, which were significantly lower than those in the SD group (78.39 and 25.31 U/L, *p* < 0.05). There was no significant difference in serum ALT activity among all groups (*p* > 0.05), but it was significantly higher than that of the normal group (*p* < 0.05). The serum AST activity of mice in each administration group showed a dose-dependent decreasing trend, but the difference was significant only between the LSEP-3 group and the HSEP-3 group (*p* < 0.05). Although the AST activity of the three administration groups was higher than that of the normal group, there was no significant difference in serum AST between the HSEP-3 group and the normal group (*p* > 0.05). Chu et al. [12] found that after sleep deprivation, the macroscopic weight reduction, gross morphological changes, and the significant increase of ALT and AST activities in the liver of rats indicated that sleep deprivation could lead to liver injury, which was similar to the results of our study. These results indicated that SEP-3 administration could effectively inhibit the abnormal changes of serum ALT and AST levels in mice caused by sleep deprivation, suggesting that SEP-3 had a repair effect on liver injury; HSEP-3 especially had a strong repair ability.

#### 2.1.3. Effects of SEP-3 on Serum GC and ACTH Levels in Sleep-Deprived Mice

To explore whether liver injury caused by sleep deprivation causes abnormal expression of GC and ACTH, we detected the serum GC and ACTH contents of mice at ZT6, ZT12, ZT18, and ZT24 time points, and the results are shown in Figure 3. At all time points, the serum GC and ACTH expression levels of SD group were the highest, the normal group was the lowest (except for HSEP-3 group in ZT6-ZT12), and the rest of the groups were between the two groups. The difference between the SD group and the normal group was significant (*p* < 0.05). Between ZT6-ZT12, the contents of GC and ACTH in the serum of mice in the normal group showed an increasing trend, but the contents of GC and ACTH in the serum of mice in the model group showed a decreasing trend. The contents of GC and ACTH in the serum of mice in the LSEP-3 and MSEP-3 groups increased slowly, which was significantly different from that in the normal group (*p* < 0.05). The content of ACTH in HSEP-3 group showed a downward trend, contrary to that in normal group, but the contents of GC and ACTH in serum of the HSEP-3 group were similar to that in the normal group, and there was no significant difference (*p* > 0.05). In the other time periods, the changes of GC and ACTH contents in serum of mice in each group were basically consistent. Compared with the SD group, the contents of GC and ACTH in serum of the administration group were decreased in a dose-dependent manner, and there was a significant difference between the HSEP-3 group and the SD group (*p* < 0.05), but no significant difference between the HSEP-3 group and the normal group (*p* > 0.05). 

### 2.2. Molecular Mechanism of SEP-3 on Liver Injury Repair in Sleep-Deprived Mice

#### 2.2.1. Effect of SEP-3 on Expression of Apoptosis-Related Proteins in Liver Tissue of Sleep-Deprived Mice

To explore the repair mechanism of SEP-3 on SDM liver injury, the expression levels of liver apoptosis-related proteins Bax, Cleaved caspase-3, and Bcl-2 were detected in this study, and the results were shown in Figure 4. Compared with the normal group, the expression levels of Bax and Cleaved caspase-3 protein in liver tissue of the SD group and each administration group were significantly increased (*p* < 0.05), and the expression level of Bcl-2 protein was significantly decreased (*p* < 0.05). Compared with the SD group, the expression levels of Bax and Cleaved caspase-3 protein in the livers of mice in the administration group were decreased in a dose-dependent manner, the expression level of Bcl-2 protein was increased in a dose-dependent manner, and the differences were significant (*p* < 0.05). The expression of Bax and Cleaved caspase-3 protein in mouse liver was negatively correlated with the dose (*p* < 0.05), and the expression of Bcl-2 protein in mouse liver was positively correlated with the dose. However, Bcl-2 protein expression was significantly different between the LSEP-3 group and the HSEP-3 group (*p* < 0.05). Bcl-2 is an anti-apoptotic protein, mainly distributed on both sides of the inner and outer membrane of mitochondria, and plays a key role in promoting cell survival and inhibiting cell apoptosis [13]. Bax is a pro-apoptotic protein, which is mainly distributed in the cytoplasm and partially adsorbed on the surface of mitochondria. It can not only change the structure of the mitochondrial membrane and affect the permeability of the mitochondrial membrane, but also induce the release of cytochrome C and activate Caspase-3 protease, thereby causing apoptosis [14]. These results indicate that SEP-3 can inhibit the apoptosis of mouse hepatocytes induced by sleep deprivation by upregulating the level of while Bcl-2, downregulating the level of Bax, and inhibiting the activity of Caspase-3 protease.

#### 2.2.2. Effect of SEP-3 on Wnt/β-Catenin Pathway-Related Protein Expression in Liver Tissue of Sleep-Deprived Mice

To further explore the repair mechanism of SEP-3 on SDM hepatocyte injury, the expression levels of Wnt/β-Catenin-pathway-related proteins β-Catenin and c-Myc in liver tissue were detected in this paper, and the results are shown in Figure 5. Compared with the normal group, the expression levels of β-Catenin and c-Myc protein in liver tissue of the SD group and each administration group were significantly decreased (*p* < 0.05), and the differences among groups were also significant (*p* < 0.05). The protein expressions of β-Catenin and c-Myc in the liver of mice in all administration groups were increased in a dose-dependent manner, and were significantly higher than those in the SD group (*p* < 0.05). β-Catenin is a biomarker to detect whether the Wnt/β-Catenin pathway is activated. Classical Wnt pathway signaling molecules can make β-catenin accumulate in the cytoplasm and enter the nucleus, regulate the expression of Wnt target genes, and promote cell proliferation and migration by blocking the degradation pathway of β-catenin [15]. The expression of c-Myc is regulated by the Wnt/β-Catenin pathway, which can be involved in regulating the expression of related genes in physiological processes such as cell growth and cell metabolism to promote cell growth and proliferation [16]. These results suggest that SEP-3 can promote the proliferation and migration of hepatocytes by activating β-Catenin and c-Myc proteins of the Wnt/β-Catenin pathway in SDM liver, thus achieving the repair of the liver injury. It has been reported that blue-mussel-derived peptides PIISVYWK and FSVVPSPK can promote the osteogenic proliferation and differentiation of hBMMSCs by activating the classical Wnt/β-Catenin signaling pathway [17].

#### 2.2.3. Regulation of SEP-3 on Core Genes of Liver Clock in Sleep-Deprived Mice

In order to explore the effect of sleep deprivation on biological rhythm and the regulatory effect of SEP-3 on biological rhythm, RT-PCR was used to detect the relative expression levels of Clock genes Bmal1, clock, Per1, Cry1, Rev-erbα, and Rorα in mouse liver at ZT6, ZT12, ZT18, and ZT24 time points. The results are as shown in Figure 6, Figure 7 and Figure 8. mRNA transcription levels of each clock gene in liver tissue of normal mice tested in this study were different at four time points, and the differences were significant (*p* < 0.05). In conclusion, the mRNA expressions of Bmal1, Clock, Per1, Cry1, Rev-erbα, and Rorα showed their own rhythm; that is, the expressions of Bmal1 mRNA, Cry1 mRNA, and Rorα mRNA showed a trend of increasing first and then decreasing. The expressions of Clock mRNA, Per1 mRNA, and Rev-erbα mRNA decreased first, then increased, and then decreased. The expressions of Bmal1, Clock, Per1, Cry1, Rev-erbα, and Rorα mRNA in SDM liver tissue were also rhythmic. The expressions of Bmal1 mRNA, Clock mRNA, Rev-erbα mRNA, and Rorα mRNA increased first, then decreased, and then increased, while the expression of Per1 mRNA increased first, then decreased, and then slowly increased. The change trend of Cry1 mRNA expression over time was the same as that of the normal group. However, the normal group reached the peak at ZT18, while the SD group reached the peak at ZT12. These results indicate that the mRNA transcription levels of bell genes in liver tissues of the normal group and SD mice show different rhythmicity or different time phases of rhythmic expression. Under normal conditions, the transcription levels of Clock genes Bmal1, Clock, Per1, Cry1, Rev-erbα, and Rorα are rhythmically stable in phase. Once the clock is disturbed, the phase and oscillation period of clock genes and their proteins will change [18]. In this study, it was found that sleep deprivation caused significant changes in the phase and oscillation period of the mRNA expressions of circadian Clock genes Bmal1, Clock, Per1, Cry1, Rev-erbα, and Rorα in the liver of mice, indicating that sleep deprivation indeed caused the disturbance of biological rhythm in mice. Zhang et al. found that the expression of liver clock gene Cry1 was significantly up-regulated in mice with insulin resistance [19]. Machicao F et al. found that Cry2 can promote the storage of triglycerides and limit the production of glucose in liver energy metabolism [20]. Doi et al. confirmed that when the CLOCK gene was mutated in mouse liver, liver glycogen synthesis was affected and insulin resistance was shown to some extent [21]. Zhou et al. found that the knockdown CLOCK or BMAL1 gene in mouse liver cells by siRNAs technology can also induce insulin resistance in liver [22]. Some scholars have found that specific knockout of BMAL1 gene in mouse liver can cause fatty liver and insulin resistance [23]. It can be seen that the normal expression of CLOCK, BMAL1, and other clock genes in the liver is an important link in regulating liver glycogen synthesis and liver sensitivity to insulin. Abnormal expression of liver clock genes may be associated with liver damage. After SEP-3 intervention, the phase and oscillation period of each clock gene expression in the liver of mice gradually tended to the normal group with the increase of drug concentration, indicating that SEP-3 can regulate the disturbance of liver biorhythm caused by sleep deprivation in mice, suggesting that SEP-3 may repair the liver injury caused by sleep deprivation by regulating the disturbance of biorhythm.

## 3. Discussion

The effects of skipjack oligopeptide SEP-3 on liver injury in SDM and its mechanism were studied. Liver index, serum ALT and AST contents can reflect the degree of liver injury [24]. The above indexes of mice in each group showed a dose-dependent trend toward the normal group, and there was no significant difference in liver index and serum AST content between the HSEP-3 group and the normal group (*p* > 0.05). According to modern medicine, GC and ACTH in vivo are mainly inactivated, degraded, and excreted by the liver [25], so the contents of GC and ACTH in serum can also reflect the degree of liver injury to a certain extent. The concentrations of GC and ACTH in serum of mice in four time periods were measured, and it was found that they also tended to be normal in a dose-dependent manner. Some studies have shown that the hypothalamic-pituitary-adrenal (HPA) axis can be excessively hyperactive under continuous stress, resulting in excessive ACTH and GC [26]. The experimental results in this paper support this conclusion. It has been found that stress can lead to HPA axis disorder, which in turn leads to increased GC and catecholamine release. The increase of GC level will promote gluconeogenesis in liver, inhibit glucose metabolism, inhibit glucose uptake in adipocytes and skeletal muscle, promote lipolysis in adipocytes, inhibit insulin secretion, cause insulin resistance and inflammation to affect glucose metabolism and guide the occurrence of metabolic disease diabetes [27]. Diabetes may promote inflammation and fibrosis through the increase of mitochondrial oxidative stress mediated by adipokines, thereby leading to liver injury [28]. These results indicated that SEP-3 could reduce the elevation of serum GC and ACTH induced by sleep deprivation in mice, suggesting that SEP-3 had a reversal effect on liver injury, especially at high dose. Dinel et al. [29] found that fish hydrolytic peptide reduced the plasma corticosterone level in mice under acute stress, which was similar to our study. The above results indicate that SEP-3 can repair liver injury in SDM.

In this paper, SEP-3 regulates hepatocyte apoptosis and the Wnt/β-Catenin pathway to explore the repair mechanism of SDM liver injury. The results showed that the expression levels of Bax and Cleaved caspase-3 were significantly increased and the expression level of Bcd-2 was significantly decreased in liver tissue after sleep deprivation. These results indicated that sleep deprivation induced hepatocyte apoptosis by increasing the level of pro-apoptotic protein Bax, and then caused liver injury. After SEP-3 intervention, the expression of Bax and Cleaved caspase-3 protein in liver tissue of mice decreased in a dose-dependent manner, while the expression of Bcl-2 protein increased in a dose-dependent manner. SEP-3 can inhibit the apoptosis of mouse hepatocytes induced by sleep deprivation by upregulating the level of BcI-2, downregulating the level of Bax, and inhibiting the activity of Caspase-3 protease. Krajewsik [30] found that the increase of anti-apoptotic protein Bcd-2 was conducive to cell survival under adverse conditions, while the increase of pro-apoptotic protein Bax could induce cell death, which was consistent with our results. Meanwhile, the expression levels of Wnt/β-catenin-pathway-related proteins tended to be normal in a dose-dependent manner. β-Catenin is a biomarker to detect whether the Wnt/β-Catenin pathway is activated. Classical signaling molecules of the Wnt pathway accumulate in the cytoplasm and enter the nucleus by blocking the degradation pathway of -catenin to regulate the expression of Wnt target genes, promoting cell proliferation and migration [12]. The expression of c-Myc is regulated by the Wnt/β-catenin pathway and can be involved in regulating the expression of related genes in physiological processes such as cell growth and cell metabolism to promote cell growth and proliferation [17]. These results indicate that SEP-3 can promote the proliferation and migration of hepatocytes by activating β-Catenin and c-Myc proteins of the Wnt/β-catenin pathway in SDM liver, so as to realize the repair of liver injury.

The relationship between sleep deprivation and liver injury, biological rhythm regulation, and liver injury repair in mice was also explored. Biological rhythm is a physiological phenomenon formed by organisms to adapt to the changes of their external environment with a cycle of about 24 h. Functions such as liver detoxification, expression and activity regulation of nuclear receptors, apoptosis, and division of hepatocytes, DNA repair, and metabolism of lipids, amino acids, and carbohydrates are all regulated by biological rhythms [31]. Sleep deprivation can lead to significant abnormal changes in the expression levels of multiple core clock genes in the liver [6,22], which may affect various physiological functions of the liver. Therefore, to explore the regulatory effect of SEP-3 on liver biorhythm in mice is of great significance for the repair of liver injury in sleep-deprived mice. The mRNA expression levels of clock-related genes in mouse liver tissues at four time points were measured. The results showed that with the increase of drug concentration, the phase and oscillation period of the mRNA expressions of Bal1, Clock, Per1, Cry1, Rev-erba, and Rora in the liver of mice gradually tended to the normal group, indicating that SEP-3 could regulate the disturbance of circadian rhythm in the liver of mice caused by sleep deprivation. These results suggest that SEP-3 may repair liver injury induced by sleep deprivation in mice by regulating the circadian rhythm of hormone disorder.

## 4. Materials and Methods

### 4.1. Animals, Materials and Reagents

SPF C57BL/6 male mice (aged 8 weeks, weighing 20–25 g, the number of samples was 120) were purchased from Hangzhou Ziyuan Laboratory Animal Technology Co., Ltd., Hangzhou, China.

The horizontal platform used for sleep deprivation modeling was customized by the research group according to the experimental requirements [32]. Specific design scheme: 18 platforms with a diameter of 2.5 cm and a height of 6 cm (fixed with threaded knobs) were placed in the rat cage, and the platform spacing was 3.5 cm. The water injection height in the cage was about 1 cm lower than the top of the platform, and the water temperature was kept at 24 ± 1 °C. The water inside the cage was always clean. The mice were free to move, eat, and drink on the platform during the experiment.

Bonito oligopeptide SEP-3 sequence Leu-Leu-Phe-Thr-Thr-Gln, purity ≥ 95% (synthesized by Shanghai Botai Biotechnology Co., LTD., Shanghai, China); ALT and AST kits (Jiancheng Bioengineering Institute); eosin dyeing solution and hematoxylin dyeing solution (Soleippo); ACTH and GC (Shanghai Enzyme-Linked Biology Co., LTD., Shanghai, China); reverse transcription kit, AceQ Qpcr SYBR Green Master Mix (Yuanxin Biological Company, Shanghai, China); Sds-page gel preparation kit, RIPA total protein lysate, BCA protein concentration determination kit, ECL chemiluminescence detection kit (ASPEN, South Africa); 0.45 μm PVDF membrane (Millipore, Burlington, MA, USA); Kodak Medical X-ray Glue (Kodak, Rochester, NY, USA) were all used in the experiment; the other reagents were imported or domestic analysis of pure.

### 4.2. Instruments and Equipment

ASP300S automatic tissue dehydrator, HistoCore Arcadia H Leica ASP Leica embedding machine, NANOCUT automatic semi-thin paraffin slicer, all from Leica, Germany; Dyy-6c electrophoresis apparatus, Beijing Liuyi Instrument Factory; Tgl-16 refrigerated centrifuge, Hunan Xiangyi Laboratory Instrument Development Co., LTD. (Changsha, China); Spark Multi-function microplate reader, Tecan, Switzerland; 5810R table top high speed refrigerated centrifuge, Eppendorf, Germany were all used in the experiment.

### 4.3. Methods

#### 4.3.1. Mouse Feeding

The mice were kept in the SPF animal room of Zhejiang Ocean University, with a temperature of about 24 °C, humidity of 55 ± 5%, and a 12 h light/dark cycle (light time from 8:00 to 20:00). All experiments were in accordance with the ethical standards of the Laboratory Animal Ethics Committee of Zhejiang Ocean University (Experimental Ethics Approval No. 2021028).

#### 4.3.2. Establishment of Sleep Deprivation Mouse Model and Administration Method

After adaptive feeding for one week, mice were randomly divided into 5 groups, with 24 mice in each group: LSEP-3, MSEP-3, and HSEP-3 groups were administered with 20 mg/kg, 40 mg/kg, and 80 mg/kg oligopeptide SEP-3 by gavage, respectively. The normal group (routine feeding, no model) and the sleep deprivation group (SD group) were administered with the same volume of normal saline. The time of gavage in each group was 8 am every day for 10 days. From the 7th day of drug administration, mice in the normal group were still routinely fed, while mice in the other four groups were fed in the modified multi-platform water environment and subjected to sleep deprivation for 72 h [33].

The time of turning on the light at 8:00 am on the 10th day was taken as the Zeitgeber time, denoted as ZT0. At the time points of ZT6 (14:00), ZT12 (20:00), ZT18 (2:00), and ZT24 (8:00), 6 mice in each group were randomly selected by eyeball harvesting and blood sampling. Centrifugation was performed at 4300× *g* for 10 min, and the supernatant was removed. Immediately after blood collection, the mice were sacrificed by cervical dislocation method and dissected immediately. The liver was quickly removed, rinsed with PBS buffer, and then dried with filter paper. The liver organ index was successively weighed and recorded.
Organ index = (organ weight mg/mouse weight g) × 10

Serum and liver samples of all mice were separated and stored in −80 °C refrigerator for later use. Each mouse was weighed and recorded before being sacrificed.

#### 4.3.3. Detection of ALT, AST, GC, and ACTH in Mouse Serum

Mouse serum frozen at −80 °C was thawed at 4 °C. The contents of ALT and AST in serum of ZT6 mice and the contents of GC and ACTH in serum of ZT6, ZT12, ZT18, and ZT24 were determined by kit.

#### 4.3.4. Detection of Apoptosis and Wnt/β-Catenin Pathway Related Protein Expression in Mouse Liver Tissue

##### Extraction and Concentration Determination of Total Protein from Liver Tissue

An appropriate amount of liver tissue was weighed and rinsed with pre-cooled PBS buffer for 2–3 times to remove blood stains, then cut into small pieces and placed in a homogenizer. An amount of 10 times the tissue volume of histone extraction reagent (a protease inhibitor was added within a few minutes before use, so that its working concentration was 1*); the ice bath was thoroughly homogenized. The homogenate was transferred to the centrifuge tube and oscillate. The ice bath was set for 30 min, during which the pipette was used to blow repeatedly to ensure that the homogenate was completely cracked. This was centrifuged at 4 °C at 12,000 r/min for 5 min and the supernatant was collected, which was the total protein solution. A BCA protein concentration assay kit was used to detect the protein concentration of samples [34].

##### SDS-PAGE Electrophoresis and Transmembrane

The protein loading concentration of the sample was adjusted to 4 μg/μL, and appropriate amount of 5× protein loading buffer was added and heated in a boiling water bath for 5 min to denature the protein. Separate glue of 8%, 10%, 12%, and concentrate glue of 5% were prepared, respectively. After adding TEMED, the glue was shaken well immediately. These were then placed into the electrophoresis tank, and the electrophoresis buffer was added, with 10 μL of the sample put into the sample hole. The concentrated glue voltage was 80 V, the separation glue voltage was 120 V, and the constant pressure electrophoresis was carried out until the bromophenol blue reached the lower edge of the glue plate. According to the molecular weight of the reference protein and the target protein, the required separation glue was cut for the transmembrane. The transfer filter paper and PVDF membrane were prepared. The PVDF membrane was activated with methanol for 3 min before use. The film transfer “sandwich” structure was placed in accordance with the direction of the positive and negative terminals. From the positive terminal to the negative terminal, the film transfer sponge, 3 layers of filter paper, PVDF film, glue, 3 layers of filter paper, and the film transfer sponge were successively placed. The bubbles in each layer should be removed during the placement process. Then, the membrane was transferred at a constant flow of 300 mA, and the transfer time was adjusted according to the molecular weight of the protein. Table 1 shows the electrophoresis and membrane transfer conditions of each target protein.

##### Antibody Incubation

After the membrane transfer was completed, the membrane was washed with TBST to remove the transfer liquid on the membrane, and the sealing liquid (5% skim milk) was added and closed at room temperature for 1 h. TBST washed off the blocking solution and the primary antibody diluted with the primary antibody diluent was added and incubated overnight by shock at 4 °C. The second antibody, diluted by the second antibody diluent, was added and incubated at room temperature for 30 min. Then, the TBST was added and washed by shock at room temperature for four times for 5 min each time. Table 2 lists the antibody information and dilution methods.

#### 4.3.5. Determination of mRNA Expression Levels of Clock-Related Genes in Mouse Liver Tissue

Total RNA was extracted from liver tissue samples using a total tissue RNA extraction kit, and cDNA was extracted and reverse transcribed using a reverse transcription kit. PCR technology was used to detect the expression of related genes, and GAPDH was taken as the target gene, and the relative mRNA expression of target genes was calculated by the 2-△△Ct method [35]. The procedure was as follows: total RNA extraction from mouse liver → reverse transcription and cDNA synthesis and detection → fluorescence quantitative PCR detection. The primer sequences of each gene are shown in Table 3.

### 4.4. Data Analysis and Processing

SPSS Statistics 25 software was used to perform one-way ANOVA on the experimental data, and Duncan’s test was used for multiple comparison analysis. The experimental results were expressed as mean ± standard deviation (mean ± SD). *p* < 0.05 indicates that the data are significantly different. Origin 2018 software was used for mapping.

## 5. Conclusions

This study showed that SEP-3 inhibited hepatocyte apoptosis and promoted hepatocyte proliferation and migration by regulating the expression of apoptosis-related proteins and Wnt/β-Catenin pathway-related proteins. At the same time, SEP-3 could improve the disturbance of liver biological rhythm caused by sleep deprivation in mice. These results suggest that it may repair the hepatocyte damage caused by sleep deprivation by regulating the disorder of the biorhythm.

## Figures and Tables

**Figure 1 marinedrugs-21-00139-f001:**
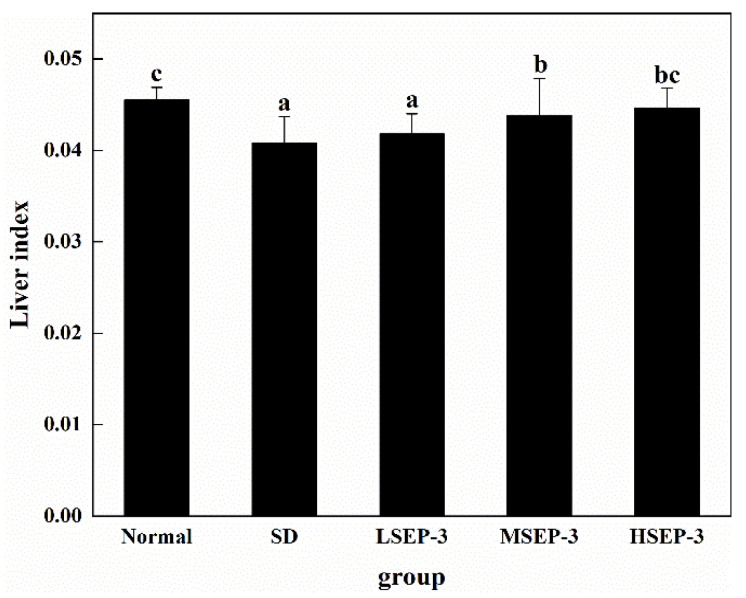
Effect of different doses of SEP-3 on the mouse liver index. Notes: the same letters in the figure indicate no significant difference between the two groups (*p* > 0.05); different letters indicate significant differences between the two groups (*p* < 0.05).

**Figure 2 marinedrugs-21-00139-f002:**
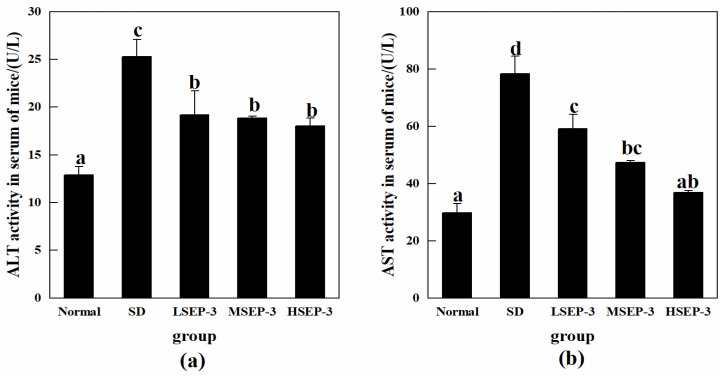
Serum ALT and AST activity of mice in each group. Notes: (**a**) shows the serum ALT content of each group, and (**b**) shows the serum AST content of each group; the same letters in the figure indicate no significant difference between the two groups (*p* > 0.05); different letters indicate significant differences between the two groups (*p* < 0.05).

**Figure 3 marinedrugs-21-00139-f003:**
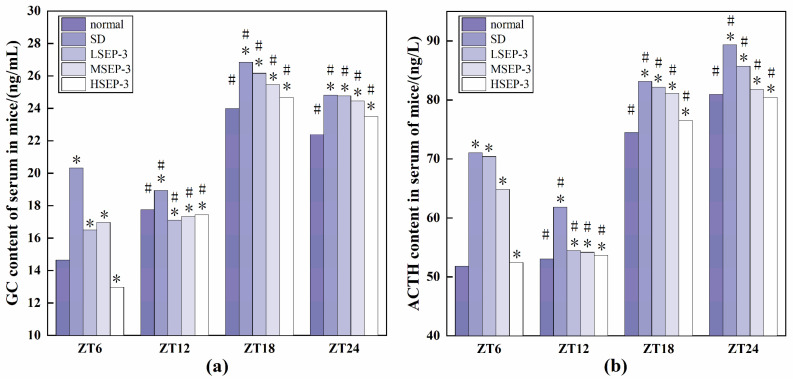
GC and ACTH content of serum in mice. Notes: (**a**) shows the serum GC content of each group, and (**b**) shows the serum ACTH content of each group; # suggests that each group index was significantly different compared with 6 h before (*p* < 0.05); * indicates a significant difference compared to the normal group at the same time (*p* < 0.05).

**Figure 4 marinedrugs-21-00139-f004:**
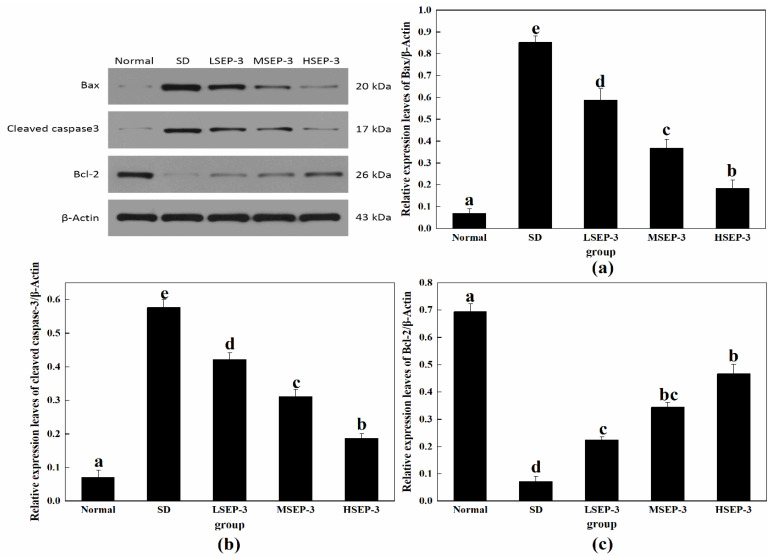
Expression of apoptosis-related proteins in liver tissues of mice. Notes: (**a**) shows the hepatic Bax expression level in each group, (**b**) is the hepatic Cleaved caspase-3 expression level in each group, and (**c**) is the hepatic Bcl-2 expression level in each group; the same letters in the figure indicate no significant difference between the two groups (*p* > 0.05); different letters indicate significant differences between the two groups (*p* < 0.05).

**Figure 5 marinedrugs-21-00139-f005:**
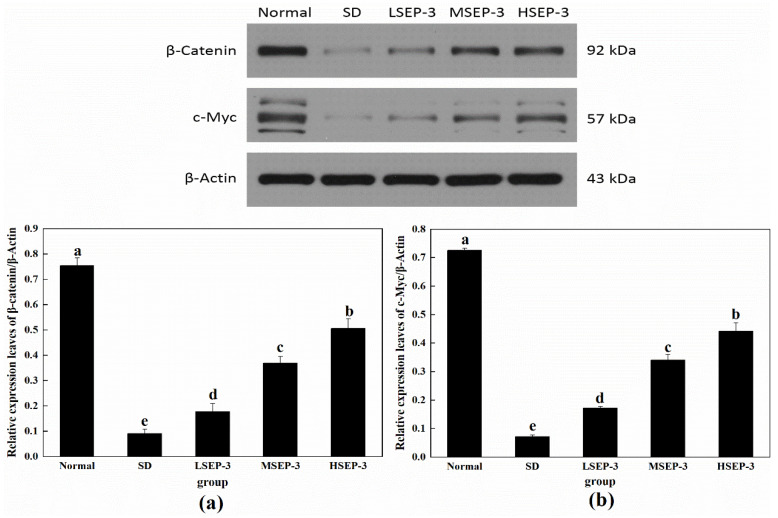
Expression of Wnt/β-Catenin pathway-related proteins in liver tissues of mice. Notes: (**a**) shows the hepatic expression level of β-Catenin in each group, and (**b**) shows the hepatic c-Myc expression level of each group;the same letters in the figure indicate no significant difference between the two groups (*p* > 0.05); different letters indicate significant differences between the two groups (*p* < 0.05).

**Figure 6 marinedrugs-21-00139-f006:**
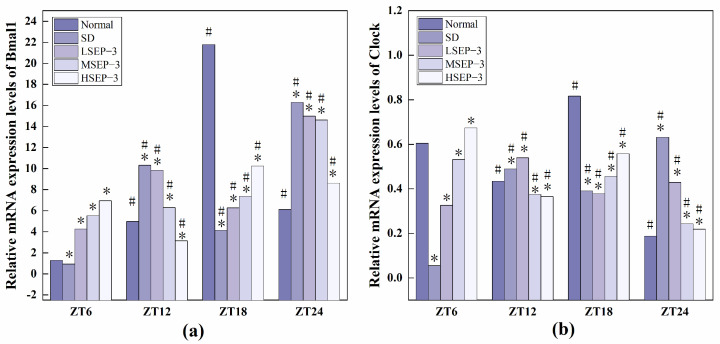
Expression levels of Clock genes Bmal1 and Clock in mouse liver tissue. Notes: (**a**) shows the expression level of Bmal 1 in each group, and (**b**) is the expression level of liver clock gene in each group; # suggests that each group index was significantly different compared with 6 h before (*p* < 0.05); * indicates a significant difference compared to the normal group at the same time (*p* < 0.05).

**Figure 7 marinedrugs-21-00139-f007:**
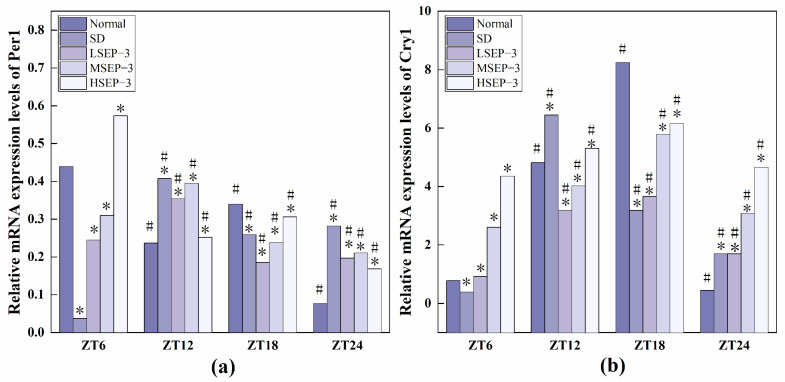
Expression levels of Clock genes Per1 and Cry1 in mouse liver tissue. Notes: (**a**) shows the expression level of the liver circadian clock gene Per 1 in each group, and (**b**) shows the expression level of Cry 1 in each group; # Suggests that each group index was significantly different compared with 6 h before (*p* < 0.05); * indicates a significant difference compared to the normal group at the same time (*p* < 0.05).

**Figure 8 marinedrugs-21-00139-f008:**
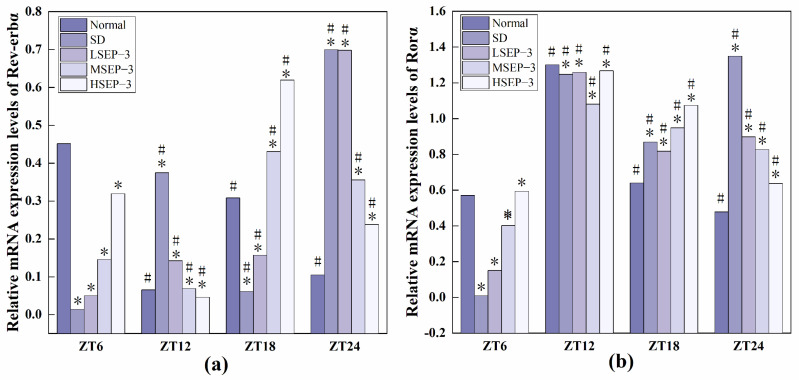
Expression levels of Clock genes Rev-erbα and Rorα in mouse liver tissue. Notes: (**a**) shows the expression level of Rev-erb α, liver clock gene in each group, and (**b**) shows the expression level of Ror α, liver circadian clock gene in each group;# Suggests that each group index was significantly different compared with 6 h before (*p* < 0.05); * indicates a significant difference compared to the normal group at the same time (*p* < 0.05).

**Table 1 marinedrugs-21-00139-t001:** Electrophoresis and membrane transfer conditions of each target protein.

Protein Name	Molecular Weight (kDa)	Separation Glue Concentration	Transfer Time (300 mA)
β-Actin	43	10%	90 min
Bax	20	12%	60 min
Cleaved caspase-3	17	10%	60 min
Bcl-2	26	10%	60 min
β-Catenin	92	8%	90 min
c-Myc	57	10%	90 min

**Table 2 marinedrugs-21-00139-t002:** Antibody information and dilution method.

Title	Origin Species	Dilution Method	Dilution Ratio
β-Actin	Rabbit	5% evaporated milk	1:10,000
Bax	Rabbit	5% BSA	1:2000
Cleaved caspase-3	Rabbit	5% evaporated milk	1:500
Bcl-2	Rabbit	5% evaporated milk	1:1000
β-Catenin	Rabbit	5% BSA	1:3000
c-Myc	Rabbit	5% evaporated milk	1:1000
HRP-Goat anti Rabbit	HRP-Goat anti Rabbit	5% evaporated milk	1:10,000

**Table 3 marinedrugs-21-00139-t003:** Oligonucleotide primers used in quantitative real-time polymerase chain reaction.

Gene	Forward Sequence (5′-3′)	Reverse Sequence (5′-3′)	PCR Products (bp)
GAPDH	AAGAAGGTGGTGAAGCAGG	GAAGGTGGAAGAGTGGGAGT	111
Bmal1	AACGGGGAAATACGGGTGA	CCTGTGGTAGATACGCCAAAAT	112
Clock	GAGATTCCATCAACACCACCA	GCCATTTTATTTAGGAGACCCA	142
Per1	TTTTGGGGCCGCTTACAG	GGGGCAGTTTCCTATTGGTTG	103
Cry1	CTGATGTATTTCCCAGGCTTTT	GCTGTCCGCCATTGAGTTCTA	195
Rorα	TGGCTTCAGGAAAAGGTAAAA	AGTCGCACAATGTCTGGGTATA	200
Rex-erbα	GCAAGGCAACACCAAGAATG	GTGCTGAGAAAGGTCACGGA	178

## Data Availability

Not applicable.

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
