# Peer review of "Repair and Mechanism of Oligopeptide SEP-3 on Oxidative Stress Liver Injury Induced by Sleep Deprivation in Mice"

_marinedrugs, 2023, doi:10.3390/md21030139_

Round 1
Reviewer 1 Report
The manuscript investigated the effect of bonito oligopeptide SEP-3 on the repair of liver damage and regulation of liver biorhythm in Sleep deprived mice.
My comments:
1. Why were three doses of the oligopeptide chosen? What is its LD50? Did the authors investigate doses of 100 mg/kg or more? Since the work clearly shows that with an increase in the dose, the therapeutic effect is enhanced.
2. In paragraph 2.1.1, the abbreviations LSEP-3, MSEP-3 and HSEP-3 should be deciphered, since they are mentioned in the text for the first time.
3. Why was only one time point ZT6 taken for the study of AST and ALT?
4. In the introduction, the authors write that more women suffer from sleep disorders. Why were male mice taken for the study?
5. What time points were taken for the Western blot? Please provide original files for western blot. It is also necessary to describe in more detail the method for proteins bax, bcl-2 and Cleaved caspase-3 indicating the manufacturers of primary and secondary antibodies.
6. It is very difficult to perceive the text with the results, since the figures with graphs are not placed in the text. Please change this moment.
Text errors^
- line 47, 61 and 252 the word skipjackis indicated twice
- line 160 - white Bcl-2
- line 188 - Results As shown in FIG. 6, 7 and 8
- line 246 and 248 - Chinese character (和)
- line 273 - KRAJEWSIK[28]
- Change in discussion Wnt/p-Catenin. There are misprints for b-catenin throughout the text.
Reviewer 2 Report
The manuscript of the authors is devoted to an interesting and relevant research topic. The article is quite well written, however, in my opinion, it has a number of shortcomings.
1. It is extremely inconvenient that the figures and their description are in different places. information is poorly received.
2. in figure 1, the signature is not sufficiently informative.
3. Figure 3, 5, 6, 8 no information on statistical significance
4. It seems to me that information on dilution of antibodies should be moved to the methods section.
5. The discussion section should be slightly expanded. in fact, there is little discussion and mostly a description of the results.
Round 2
Reviewer 1 Report
1.There is a misprint in the text describing Figure 2. In line 94 for the SD group, you need to change the concentrations of ALT and AST.
2. The authors say that the mortality of female mice was higher (in response to my first remarks) than males, and at the same time write in the answer that the LD50 dose is not known. Please explain this point in more detail.
3. Images of triplets for western blot are desirable (possibly in the form of additional files).
4. Correct the name of the protein β-catenin in the discussion section of lines 302 to 314.
Author Response
Female mice died more during sleep deprivation (when untreated), while males did not die, suggesting other factors, so male mice were used. The LD50 testing of SEP-3 was not performed.
1.Related error has been changed. 2.Female mice died more during sleep deprivation (when untreated), while males did not die, suggesting other factors, so male mice were used. The LD50 testing of SEP-3 was not performed. 3.See the attachment for the picture. 4.The spelling error has been corrected.